# Towards learning principles of the brain and spiking neural networks

## Abstract

The brain, the only system with general intelligence, is a network of spiking neurons (i.e., spiking neural networks, SNNs), and several neuromorphic chips have been developed to implement SNNs to build power-efficient learning systems. Naturally, both neuroscience and machine learning (ML) scientists are attracted to SNNs' operating principles. Based on biologically plausible network simulations, we propose that spatially nonspecific top-down inputs, projected into lower-order areas from high-order areas, can enhance the brain's learning process. Our study raises the possibility that training SNNs need novel mechanisms that do not exist in conventional artificial neural networks (ANNs) including deep neural networks (DNNs).

## 1 Introduction

Spiking neural networks (SNNs) are our brain's building blocks and known to be power-efficient. With the brain's general intelligence in mind, it seems only natural to construct artificial SNNs to advance ML; see [8] for a review. However, training SNNs is challenging, and our knowledge of their exact operating principles remains insufficient; see [7] for a review. In principle, underlying mechanisms of brain's learning can help us develop learning algorithms for SNNs.

Spike-time dependent plasticity (STDP) has been thought to underlie the brain's learning capabilities [2]. Specifically, STDP can strengthen synaptic connections if presynaptic neurons fire earlier than postsynaptic neurons but can weaken them if the postsynaptic neurons fire earlier. That is, STDP can selectively promote causal connections, which is necessary (while not sufficient) for learning. In the brain, however, neurons are intricately connected, and a line of studies suggested that some top-down signals may not be target-specific [1, 5, 9]. Then, how does STDP selectively modulate synaptic connections' strengths in such complex networks?

To gain more insights, we studied STDP connections and its evolution with and without nonspecific top-down connections, using biologically plausible network models. Specifically, we considered two inhomogeneous cell assemblies in a low-order area and a single assembly in a higher order area. One low-order assembly is stimulated and expected to make stronger connections to the high-order assembly. Our simulation results suggest that nonspecific top-down connections make bottom-up connections's learning more efficient, raising the possibility that SNNs' learning process may require additional mechanisms which do not exist in traditional DNNs.

Submitted to 33rd Conference on Neural Information Processing Systems (NeurIPS 2019). Do not distribute.

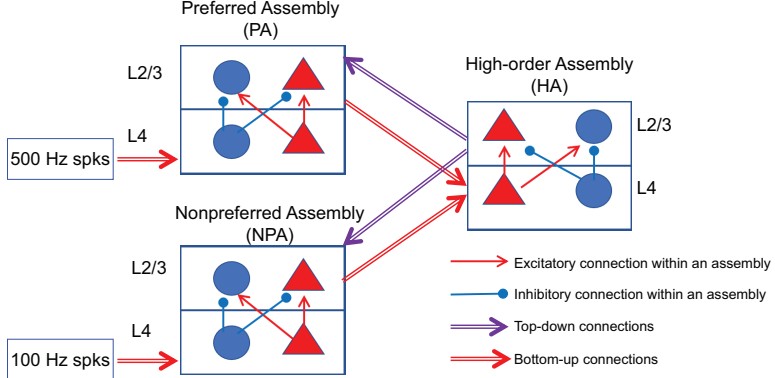

Figure 1: Schematics of the model, which consists of three assemblies (PA, NPA and HA). In superficial (L2/3) and granular (L4) layers, Pyr (red rectangles) and PV (blue circles) interact with one another. PA and NPA represent the low-order area, and they receive disparate external inputs. HA represents the high-order area. All these three assemblies interact with layer-specific top-down and bottom-up connections.

## 2  Results and Discussion

As shown in Fig. 1, our model consists of two cell assemblies in a low-order area and a single assembly in a high-order area (see Section 3 for details). The two assemblies in the low-order area project to the high-order area (HA) that projects back to them via nonspecific top-down connections. Initially, the bottom-connections from both low-order area assemblies are identical and strong enough to innervate the high-order area assembly (HA), when the low-order area assemblies generate sufficiently strong outputs. With these bottom-up connections, when any of the low-order area assemblies becomes active, HA will fire. That is, the selected low-order assembly and HA will fire together, and we assume that these two connections need to be selectively strengthened, as Hebbian learning rule proposed [3]. As spike-time-dependent-plasticity (STDP) has been thought to underlie Hebbian learning in the brain [6], we implement STDP in all bottom-up connections in the model and look into nonspecific top-down inputs' contribution to bottom-up connections' learning.

In our model, we randomly choose a preferred assembly (PA) to which we introduce 500 Hz external inputs, but a non-preferred assembly (NPA) receives 100 Hz. That is, the bottom-up connections from PA to HA are expected to grow according to Hebbian learning rule. For the sake of brevity, bottom-up connections from PA and NPA will be referred to as Conn_PA and Conn_NPA, respectively, hereafter. Due to the observations that inter-area connections are layer-specific, all three assemblies consist of superficial (L2/3) and granular (L4) layers, in which excitatory and inhibitory neurons interact with one another via randomly established connections (Table 1). We refer to excitatory neurons as 'Pyr' neurons, as most excitatory neurons are shaped like pyramids, and inhibitory neurons as 'PV' neurons, as most common molecular markers of inhibitory neurons are parvalbumin (PV).

We simulate the network for 20 seconds (s) to estimate how Conn_NPA and Conn_PA evolve over time with and without nonspecific top-down inputs onto both NPA and PA (Fig. 1). Fig. 2 A, B and C show the spikes from PA, NPA and HA for the first 3 s; Pyr and PV neurons are shown in red and blue, respectively. As shown in the figures, Pyr neurons in all assemblies fire synchronously several times. We note that the synchronous activity appears first in PA, which receives 500 Hz afferent inputs, and it subsequently appears in L4 neurons of HA and L2/3 neurons of NPA (Fig. 2D). This pattern can be readily explained by the patterns of bottom-up and top-down connections in the model. More importantly, when such sequential activations occur, according to STDP rule, Conn_PA grow stronger, whereas Conn_NPA grow weaker. Indeed, this sequential activation occurs throughout the simulation. Consequently, Conn_PA grow stronger gradually, but Conn_NPA grow weaker gradually (Fig. 3A), suggesting that bottom-up connections can be selectively strengthened with

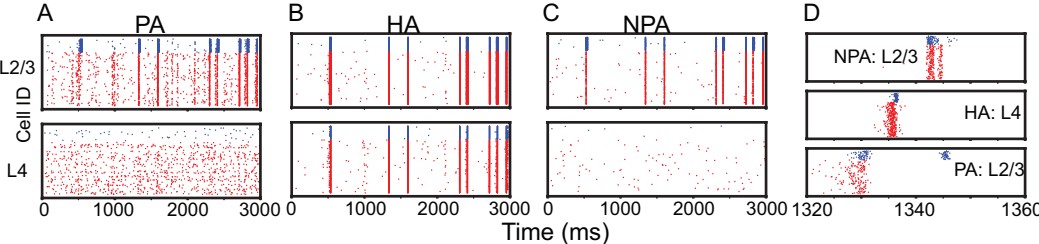

Figure 2: Raster plots of an example simulation. (A), (B), (C), spikes generated in L2/3 and L4 of PA, HA and NPA, respectively. Each dot represents a spike, and spikes from Pyr and PV neurons are shown in red and blue. For the clarity, we show them only for the first 3 seconds. (D), spikes from L2/3 of NPA and PA and those from L4 of HA, between 1320 and 1360 ms.

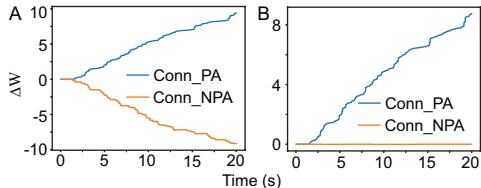

Figure 3: Time courses of bottom-up connections. (A), the mean values of bottom-up connections as a function of time. The blue and yellow lines represent the mean values of bottom-up connections from PA to HA and those from NPA to HA, respectively. The mean values are calculated every 5 ms during simulations. (B), the same as (A) but without top-down connections.

nonspecific top-down inputs by maintaining the total strengths of bottom-up connections roughly at the same level.

To further examine the functions of nonspecific top-down inputs, we repeat the simulation without top-down inputs (both to PA and NPA). When top-down connections are removed from the model, Conn_PA increase as before, but Conn_NPA remain unchanged (Fig. 3B). These results suggest that nonspecific top-down connections can reduce the strengths of undesired bottom-up connections (i.e., connections from NPA to HA in this model).

Although our simulations consider a simple learning scenario directly linked to Hebbian learning, our results suggest that feedback (i.e., top-down) connections, ignored in traditional DNNs, can make the brain's learning more efficient in two ways. First, the divergence of synaptic connection's strengths grows bigger with nonspecific feedback inputs, which can increase SNNs' learning capability. Second, more synapses can be trained in parallel, which can shorten the training times of SNNs. In the future, we will extend the model to test more realistic learning scenarios.

Finally, we note that many circuit motifs in the brain have been recently discovered (see [7] for instance), but their functions remain elusive. Based on our results, we argue that biologically plausible network models will allow us to better understand neural circuit motifs' contributions to the brain's learning and gain insights into general learning algorithms suitable for SNNs. Properly selected learning scenarios and learning algorithms' objectives would strengthen this type of research, which necessitates a collective effort between machine learning scientists and neuroscientists.

## 3 Methods

We use the peer-reviewed open-source simulation platform NEST [4]. to build the network model. In the model, all neurons are current-based leaky-integrate fire (LIF) neurons. All neu-

Table 1: Connections in the network model. Below, the connection probability and strength of each connection type are shown in the parenthesis. TD, BU and LGN represent top-down, bottom-up and LGN connections, respectively. Additionally, Pyr and PB neurons receive 1050 Hz and 1000 Hz background inputs, respectively, via 100 pA connections.

| | | Postsynaptic Neurons | | | |
| --- | --- | --- | --- | --- | --- |
| | | L2/3 Pyr | L2/3 PV | L4 Pyr | L4 PV |
| Presynaptic Neurons | L2/3 Pyr | (0.4, 40 pA) | (0.6, 40 pA) | N/A | N/A |
| | L2/3 PV | (1.0, -40 pA) | (1.0,-40 pA) | N/A | N/A |
| | L4 Pyr | (0.6, 80 pA) | (0.4, 40 pA) | (0.4, 40 pA) | (0.6, 40 pA) |
| | L4 PV | N/A | N/A | (1.0, -40 pA) | (1.0, -40 pA) |
| Across assemblies and external inputs | | | | | |
| TD to Pyr | (0.3, 15 pA) | | BU to PV | (0.3, 20 pA) | |
| TD to PV | (0.3, 20 pA) | | LGN to Pyr | (0.3, 60 pA) | |
| BU to Pyr | (0.3, 40 pA) | | LGN to PV | (0.3, 30 pA) | |

Table 2: Parameters for neurons and synaptic inputs.

| Param | Value | Param | Value |
| --- | --- | --- | --- |
| Membrane constant | 10 ms | $\tau_s \left( Pyr \rightarrow PV \right)$ | 2 ms |
| Spike threshold | -50 mV | $\tau_s \left( PV \rightarrow Pyr \right)$ | 6 ms |
| Reset potential | -65 mV | $\tau_s \left( PV \rightarrow PV \right)$ | 4.3 ms |
| Refractory Period | 2 ms | Pyr cell # | 320 |
| $\tau_s \left( Pyr \rightarrow Pyr \right)$ | 2 ms | PV cell # | 80 |

rons and synaptic connections are implemented using NEST's native models. Any parameters not specified in the table 2 are taken from the NEST package's default parameters[4].

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
