# OpenReview forum: "Towards learning principles of the brain and spiking neural networks"
_NeurIPS.cc/2019/Workshop/Neuro_AI — Submitted to Real Neurons & Hidden Units @ NeurIPS 2019_

### Official Review · AnonReviewer2 · 2019-09-26
**Nice basic idea but limited exploration of the model**

**Clarity:** 4

**Comment:**

As mentioned in my response to the technical rigor section, this study could be greatly improved by further simulations exploring parameter sensitivity, STDP rule choices, and network architecture choices. The basic idea has merit but needs to be better explored.

**Category:**

Common question to both AI & Neuro

**Clarity Comment:**

Overall it is well written and the figures are clear.

However neither of the two conclusions the authors make are clear to me: "First, the divergence of synaptic connection’s strengths grows bigger with nonspecific feedback inputs, which can increase SNNs’ learning capability. Second, more synapses can be trained in parallel, which can shorten the training times of SNNs." Elaboration or

**Evaluation:**

3: Good

**Importance:**

2: Marginally important

**Importance Comment:**

This is a clever basic idea. As far as I can tell the basic underlying mechanism is similar to the single-cell competitive STDP process proposed by Song, Miller and Abbott (2000), but at a circuit level, and loosely mapped on to cortical hierarchy. This aspect I believe is novel and interesting.

**Intersection:**

2: Low

**Intersection Comment:**

The study is straight-ahead computational neuroscience. Although the text mentions potential applications to machine learning, neuromorphic computing, and deep learning, ML-style models are not implemented. It may be that mapping these mechanisms onto ANNs is not straightforward. The two conclusions mentioned

**Rigor Comment:**

Overall the results were pretty minimal and could have been expanded to strengthen the authors' case. The results are something like "proof by example". It would be more convincing to me if the authors could explore the robustness and generality of this mechanism. How does it depend on parameter choices? What regimes will it have strongest effect and when will it break? What is the role of the separate L2/3 and L4 networks? Although they mimic the cortical anatomy, what computational functions do they perform here?

The methods section could be more elaborate to aid reproducibility. For example the STDP model is not described at all, and it is known that the implementation details can affect competition (additive vs multiplicative weight changes being one example). Also STDP simulations are notoriously sensitive to parameter choices.

**Technical Rigor:**

2: Marginally convincing

---

### Official Review · AnonReviewer3 · 2019-09-27
**Nice preliminary idea but with limited investigation/analysis of the model and results**

**Clarity:** 4

**Comment:**

The ideas and results presented in the paper are novel but as of now there doesn't seem to be enough analysis to fully support the author's claims. As mentioned in 'technical rigor' section, the paper could be greatly improved with further exploration of the model.

**Category:**

Common question to both AI & Neuro

**Clarity Comment:**

Overall the paper is well written and easy to follow

**Evaluation:**

3: Good

**Importance:**

3: Important

**Importance Comment:**

The paper provides circuitry level simulation setup of STDP process for hierarchical network structure as observed in brain which is an interesting idea and could go long way with systematic exploration.

**Intersection:**

2: Low

**Intersection Comment:**

The concepts discussed in the paper seem to have much stronger association with computational neuroscience rather than ML. Although author does mention potential applications to machine learning but it is unclear how the mechanism presented in the paper could be implemented into artificial neural nets.

**Rigor Comment:**

The result presented is nice but could have been expanded with more analysis driven by different variations to the model parameters/input regimes/network structure to better understand the mechanisms in actions. As of now the results presented seem incomplete to fully support the authors' conclusions.

**Technical Rigor:**

2: Marginally convincing

---

### Official Review · AnonReviewer1 · 2019-09-30
**Interesting observation with limitations**

**Clarity:** 4

**Comment:**

I believe this idea needs a more rigorous evaluation and better motivation. A simple search for "stdp with top down feedback" or similar turns up a multitude of similar models; the authors should clarify what's their contribution.

**Category:**

Not applicable

**Clarity Comment:**

The manuscript is easy to follow.

**Evaluation:**

2: Poor

**Importance:**

2: Marginally important

**Importance Comment:**

Albeit similar models have been explored previously, investigating the role of top-down feedback in learning could be important for understanding learning in biological neural circuitry.

**Intersection:**

3: Medium

**Intersection Comment:**

The machine learning relevance of the proposed approach is not obvious.

**Rigor Comment:**

The initial experiments provided hint at possible roles of top-down feedback in learning, however, more evidence is necessary to make significant conclusions. Neither theoretical nor intuitive justification is provided for what is observed.
In particular, the model considered is just one of many possible ones (there could be inhibitory feedback, too, for instance, and for a more realistic setting, the feedback connections should possibly be plastic, as well.)
It is not obvious to me how the results shown in fig. 3 come about. It seems like the blue curve is exactly the same in both cases, while the orange curve is just constant (zero) in the case without feedback. Since STDP normally leads to weight changes even without feedback, something must be unusual here.

**Technical Rigor:**

2: Marginally convincing

---

### Decision · Program_Chairs · 2019-10-01

**Decision:**

Reject

**Comment:**

Unfortunately, we had more submissions than we could accept and based on the review process, we have decided not to accept your submission.  Nevertheless, thank you for your submission and interest in our workshop.